# SPARK: Spatio-temporal Part-based Attention for Retargeting Cross-skeleton Motion

## Abstract

Cross-skeleton motion retargeting remains a challenging problem in computer animation, particularly when dealing with characters having significantly different skeletal structures. Existing methods often struggle to preserve motion semantics while adapting to diverse skeleton topologies. We propose a novel transformer-based approach that leverages group-based body part processing and spatio-temporal attention mechanisms. Our method organizes joints into semantic body groups and employs attention pooling to generate robust representations that capture both local joint relationships and global body dynamics. A transformer encoder models temporal dependencies across these body-part tokens, learning motion patterns invariant to specific skeletal configurations. The decoder uses cross-attention to enable fine-grained motion transfer by attending to spatial body part correspondences and temporal motion patterns. We incorporate T-pose conditioning and joint text embeddings to provide anatomical structure awareness during retargeting. Evaluation on the Mixamo dataset demonstrates particular strength in handling complex skeletal variations while maintaining motion quality and semantic consistency. We will release the code to facilitate reproducibility and future research.

## 1 Introduction

Motion retargeting, the process of transferring motion sequences between characters with different skeletal structures, is essential across gaming, virtual reality, film, and robotics industries. As digital content creation scales, studios frequently need to adapt existing motion capture data to characters with vastly different anatomical configurations. Currently, this process relies on manual intervention by skilled animators, creating a time-intensive bottleneck in production pipelines. The need for robust automated cross-skeleton motion retargeting has thus become a fundamental research challenge, promising to accelerate content creation while enabling efficient reuse of motion assets across diverse character types.

With the advancement of deep learning, numerous approaches have emerged that leverage neural networks to automate motion retargeting through an encode-decode paradigm: first encoding motion from the source skeleton, then decoding it for playback on the target skeleton. Early methods Villegas et al. (2018); Aberman et al. (2020); Lee et al. (2023) employed RNN or Graph Neural Networks (GNNs) Zhou et al. (2021) to aggregate information across different skeletal structures, but these approaches require predefined network architectures tailored to specific joint configurations, limiting their generalizability across skeletons with varying numbers of joints. To address this limitation, recent transformer-based methods Martinelli et al. (2024); Zhang et al. (2023) have been proposed to handle variable skeletal topologies through self-attention and cross-attention mechanisms Vaswani et al. (2023). However, the computational complexity of these attention operations scales quadratically with sequence length and joint count, resulting in prohibitively high computational costs and training times, particularly for long motion sequences or complex skeletal structures.

To address these limitations, we propose SPARK (**S**patio-temporal **P**art-based **A**ttention for **R**etargeting Cross-s**k**eleton Motion), a novel approach that efficiently handles skeletons with varying joint numbers while significantly reducing computational overhead. Our method introduces a semantic body part grouping strategy that organizes skeletal joints into anatomically meaningful clusters (e.g., torso, limbs, extremities), enabling consistent motion representation across diverse

skeletal topologies. Rather than applying attention across all individual joints, SPARK operates on these semantic groups, dramatically reducing the computational complexity from quadratic to linear scaling. We further enhance efficiency through a spatio-temporal cross-attention mechanism that captures both spatial relationships between body parts and temporal motion dynamics within a unified framework. Additionally, our approach incorporates a part-aware positional encoding scheme that preserves the geometric relationships within each body part while maintaining semantic correspondence across different skeletal structures. Extensive experiments demonstrate that SPARK achieves superior motion quality compared to existing methods while requiring substantially less training time and computational resources.

Our main contributions are threefold. First, we introduce a semantic body part grouping strategy that enables consistent motion representation across skeletons with varying joint numbers while reducing computational complexity from quadratic to linear scaling. Second, we propose a spatio-temporal cross-attention mechanism combined with part-aware positional encoding that effectively captures both spatial relationships between body parts and temporal motion dynamics. Third, we demonstrate through comprehensive experiments that our approach achieves superior motion retargeting quality compared to state-of-the-art methods while requiring significantly less training time and computational resources.

## 2 RELATED WORKS

### 2.1 SKELETAL VARIATION

Character skeletons can differ by topology, which we classify into three categories: Isomorphic skeletons, Homeomorphic skeletons, and Non-homeomorphic skeletons. It is difficult to perform motion retarget and motion gen with the last two skeletons. Handling homeomorphic skeletons requires methods to bridge small topology gaps, such as extra or fewer joints along a chain. Graph-based approaches like Skeleton-Aware Networks Aberman et al. (2020) reduce homeomorphic skeletons to a common "primal skeleton" representation using differentiable skeletal pooling operations, enabling retargeting between characters with different joint counts by encoding to and decoding from a shared latent space. More recent works like M-R²ET Zhang et al. (2024) extend this paradigm by learning automatic joint correspondence mappings and applying residual correction modules to maintain motion semantics and physical plausibility across homeomorphic variations. Non-homeomorphic cases involving fundamentally different skeletons are the most challenging and historically require manual intervention, with traditional methods like Yamane et al. (2010) needing paired motion data and Creature Features Seol et al. (2013) requiring manual joint mapping definitions. Recent transformer-based approaches like Martinelli et al. (2024) treat each joint's motion as an independent token using masked autoencoder training to learn implicit correspondences without paired data, while the AnyTop framework Gat et al. (2025) uses diffusion-based generation with graph-aware encoding to produce motions for arbitrary skeletal topologies. These approaches demonstrate significant progress in cross-topology motion transfer, though performance on completely unseen skeleton types may still require further investigation.

### 2.2 MOTION RETARGETING

Cross-skeleton motion retargeting is challenging because motions must be transferred between characters with different skeletal topologies and proportions while preserving semantic meaning. Early works like SAN Aberman et al. (2020) and SAME Lee et al. (2023) define Graph Neural Networks (GNNs) on skeleton graphs, enabling motions to be embedded into common latent spaces shared by homeomorphic skeletons. PAN Hu et al. (2024) follows a similar GNN approach but focuses on part-based operations. However, since these network structures are fundamentally defined based on joint numbers, their generalization to characters with significantly different structural topologies remains limited. To handle skeletons with varying joint numbers, recent approaches like R2ET Zhang et al. (2023) and MoMa Martinelli et al. (2024) employ transformers to encode different skeleton structures. While effective, the quadratic complexity of transformer attention mechanisms results in substantial computational overhead and slow training. Motion2Motion Chen et al. (2025) proposes a lightweight, training-free retargeting method based on motion retrieval. However, this approach requires explicit joint correspondence definition between skeletons and assumes the availability of motion sequences for target skeletons, which may not always be practical.

## 3 PRELIMINARY

**Attention pooling**. Attention pooling is a mechanism that aggregates variable-length sequences into fixed-size representations by learning to focus on the most relevant elements. It builds upon the attention mechanism popularized by transformers, but differs in its application: while transformers typically use attention for sequence-to-sequence modeling, attention pooling specifically targets sequence-to-fixed-size aggregation.

Given a sequence of feature vectors $\mathbf{X} = \{\mathbf{x}_1, \mathbf{x}_2, \ldots, \mathbf{x}_n\} \in \mathbb{R}^{n \times d}$, attention pooling employs the same scaled dot-product attention as transformers but with learnable query vectors. We implement attention pooling using the standard transformer attention structure, where a set of learnable query vectors $\mathbf{Q} = \{\mathbf{q}_1, \mathbf{q}_2, \ldots, \mathbf{q}_m\} \in \mathbb{R}^{m \times d}$ are initialized with uniform random values and trained to extract $m$ different aspects of the input sequence.

For each query $\mathbf{q}_i$, the attention weights are computed as:

$$\alpha_{i,j} = \frac{\exp(\mathbf{q}_i^T \mathbf{x}_j / \sqrt{d})}{\sum_{k=1}^{n} \exp(\mathbf{q}_i^T \mathbf{x}_k / \sqrt{d})} \tag{1}$$

Then the pooled representation for query $\mathbf{q}_i$ is $\mathbf{z}_i = \sum_{j=1}^{n} \alpha_{i,j} \mathbf{x}_j$. This results in a fixed-size output $\mathbf{Z} = \{\mathbf{z}_1, \mathbf{z}_2, \ldots, \mathbf{z}_m\} \in \mathbb{R}^{m \times d}$ regardless of the input sequence length $n$.

In our approach, we employ attention pooling with $m = 4$ queries for each skeleton part, allowing the model to capture four distinct motion characteristics per body part. This design choice balances expressiveness with computational efficiency, providing sufficient representational capacity while maintaining manageable model complexity.

## 4 METHOD

Given a source skeleton $\mathcal{S}_s$ with $N_s$ joints and a corresponding motion sequence $\mathcal{M}_s = \{\mathbf{p}_s^{(1)}, \mathbf{p}_s^{(2)}, \ldots, \mathbf{p}_s^{(T)}\}$, where $\mathbf{p}_s^{(t)} \in \mathbb{R}^{N_s \times 6}$ represents the 6D joint rotations Zhou et al. (2020) at time step $t$, our goal is to retarget this motion to a target skeleton $\mathcal{S}_t$ with $N_t$ joints, producing a semantically equivalent motion sequence $\mathcal{M}_t = \{\mathbf{p}_t^{(1)}, \mathbf{p}_t^{(2)}, \ldots, \mathbf{p}_t^{(T)}\}$, where $\mathbf{p}_t^{(t)} \in \mathbb{R}^{N_t \times 3}$.

The key challenge lies in the fact that $\mathcal{S}_s$ and $\mathcal{S}_t$ may have significantly different topologies, with $N_s \neq N_t$ and distinct joint connectivity patterns and different joint bone lengths. Despite these structural differences, the retargeted motion $\mathcal{M}_t$ must preserve the semantic content and naturalness of the original motion $\mathcal{M}_s$ while being anatomically plausible for the target skeleton.

Formally, we seek to learn a mapping function $f : (\mathcal{S}_s, \mathcal{M}_s, \mathcal{S}_t) \rightarrow \mathcal{M}_t$ that maintains motion semantics across different skeletal structures. This requires the model to understand the functional correspondence between body parts across skeletons and transfer motion characteristics accordingly, rather than relying on explicit joint-to-joint mappings.

In the following, we describe how we group joints in 4.1, how we design the motion encoder in 4.2, and the decoder in 4.3. Finally, we show the overall architecture of our training and testing pipeline in 4.4, followed by the training objective descriptions.

### 4.1 JOINT GROUPS

The key insight behind our approach is that human motion exhibits inherent locality: joints that are anatomically distant have minimal direct influence on each other during movement. For instance, hand movements are largely independent of foot motions, while joints within the same limb are highly correlated due to shared neural pathways and biomechanical constraints. In mathematical terms, spatially distant joints require minimal attention, whereas proximate joints within the same kinematic chain exhibit strong interdependencies that benefit from focused attention mechanisms.

Inspired by this biological principle, we partition the skeleton into six semantic body parts: torso, left leg, right leg, left arm, right arm, and head. This grouping reflects the natural organization of the human musculoskeletal system, where each group corresponds to a functionally coherent unit controlled by related neural and muscular structures.

The torso group plays a central role in our partitioning scheme, as it represents the spine—the primary structural axis that coordinates overall body movement and serves as the kinematic foundation for limb motions. So different from PAN Hu et al. (2024), in which each body group shares no joints with other groups, to capture this biomechanical hierarchy, we design overlapping connections between groups: the left and right leg groups share the hip joint with the torso, while the left arm, right arm, and head groups share the uppermost spine joint with the torso. This design ensures that the torso's coordinating influence on peripheral body parts is preserved while maintaining the locality principle within each group. We will show in the ablation study about how this design helps improve the performance of motion retargeting.

## 4.2 MOTION ENCODER

**Spatial space attention.** Our motion encoder processes each semantic body part independently through a specialized spatial attention mechanism. For a motion sequence with $T$ frames and batch size $B$, we first organize joints according to the six semantic groups defined in Section 4.1. Each joint group $g_i$ (where $i \in \{1, 2, \ldots, 6\}$) contains a variable number of joints $n_i$. To enable efficient batch processing, we pad each group to its maximum joint count $\max(n_i)$, resulting in group representations of size $\mathbb{R}^{T \times B \times \max(n_i) \times D}$, where $D$ is the feature dimension. For each joint within a group, we incorporate three types of embeddings to provide rich contextual information. (1) Positional encoding: we add learnable positional encodings $\mathbf{E}_{pos}^{(i)} \in \mathbb{R}^{\max(n_i) \times D}$ to distinguish joint positions within each group. (2) Joint name embeddings: we extract joint names from the original BVH motion files and encode them using the T5 text encoder Raffel et al. (2023). The resulting joint name embeddings $\mathbf{E}_{name}^{(i)} \in \mathbb{R}^{\max(n_i) \times D}$ capture semantic relationships between anatomically similar joints. (3) T-pose embeddings: inspired by Guo et al. (2022); Gat et al. (2025), we adopt a redundant representation for T-pose. Each joint $j$ (except the root) consists of its root-relative position $p_j \in \mathbb{R}^3$, 6D joint rotation $r_j \in \mathbb{R}^6$, linear velocity $v_j \in \mathbb{R}^3$, and foot contact label $fc_j \in \{0, 1\}$. The T-pose representation is processed through an MLP and then added to every frame-axis of the motion embedding, providing temporal context that helps the model understand the motion dynamics and phase information across the sequence. After the enhancement, we get $\mathbf{X}_{enhanced}^{(i)} = \mathbf{X}^{(i)} + \mathbf{E}_{pos}^{(i)} + \mathbf{E}_{name}^{(i)} + \mathrm{MLP}(\mathbf{tpos}^{(t)})$.

To improve the model's robustness to varying skeletal structures, we randomly mask joints in each group by setting them to zero. We then apply transformer-based attention pooling along the joint axis for each group. As described in Section 3, each group employs $m = 4$ learnable queries, enabling the extraction of four distinct motion characteristics per body part $\mathbf{Z}^{(i)} = \mathrm{AttentionPooling}(\mathbf{X}_{enhanced}^{(i)}, \mathbf{Q}^{(i)})$, where $\mathbf{Q}^{(i)} \in \mathbb{R}^{4 \times D}$ are the learnable query vectors for group $i$, and $\mathbf{Z}^{(i)} \in \mathbb{R}^{T \times B \times 4 \times D}$ represents the pooled features. Finally, we concatenate the outputs from all six groups to obtain a unified representation $\mathbf{Z} = \mathrm{Concat}([\mathbf{Z}^{(1)}, \mathbf{Z}^{(2)}, \ldots, \mathbf{Z}^{(6)}]) \in \mathbb{R}^{T \times B \times 24 \times D}$. This design yields 24 tokens ($4 \times 6$) that capture the essential motion characteristics across all semantic body parts while maintaining spatial locality within each group.

**Temporal space attention.** Following the spatial attention pooling, we process the temporal dynamics through our main transformer encoder blocks. The concatenated spatial features $\mathbf{Z} \in \mathbb{R}^{T \times B \times 24 \times D}$ from the previous stage serve as input to this temporal processing module. We employ multi-head attention mechanisms applied along the temporal axis to capture long-range temporal dependencies in the motion sequence. The temporal attention operates on each of the 24 semantic tokens independently, allowing the model to learn how motion characteristics evolve over time within each body part.

For each token position $j \in \{1, 2, \ldots, 24\}$, we extract the temporal sequence $\mathbf{z}_j = [\mathbf{z}_j^{(1)}, \mathbf{z}_j^{(2)}, \ldots, \mathbf{z}_j^{(T)}]^T \in \mathbb{R}^{T \times D}$ and apply multi-head attention:

$$\mathbf{h}_j^{(\ell)} = \mathrm{MultiHeadAttn}(\mathbf{h}_j^{(\ell-1)}) + \mathbf{h}_j^{(\ell-1)} \tag{2}$$

$$\mathbf{h}_j^{(\ell)} = \mathrm{FFN}(\mathbf{h}_j^{(\ell)}) + \mathbf{h}_j^{(\ell)} \tag{3}$$

where $\mathbf{h}_j^{(0)} = \mathbf{z}_j$, $\ell \in \{1, 2, 3, 4\}$ denotes the transformer block index, and FFN represents the feed-forward network with residual connections and layer normalization.

Our temporal encoder consists of 4 transformer blocks, providing sufficient depth to model complex temporal patterns while maintaining computational efficiency. Each block follows the standard transformer architecture with multi-head self-attention, position-wise feed-forward networks, and residual connections. The output of the temporal encoder is a refined representation $\mathbf{H} \in \mathbb{R}^{T \times B \times 24 \times D}$ that encodes both spatial motion characteristics and their temporal evolution, serving as a comprehensive motion representation for the subsequent retargeting process.

The structure of the motion encoder is shown on the left side of Figure. 1.

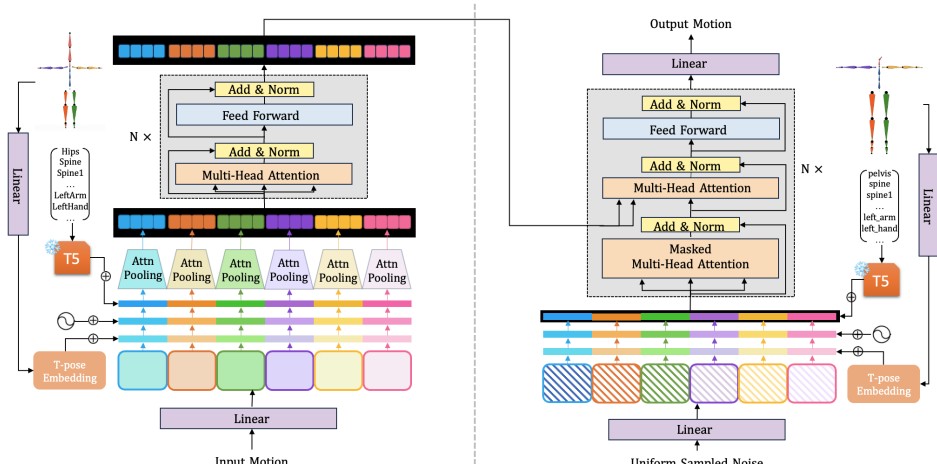

Figure 1: Our Transformer-based encoder and decoder structure. Attention pooling is only used in the encoder. The uniformly sampled noise to put into the decoder is of the same size as the output motion.

### 4.3 MOTION DECODER

The decoder transforms the encoded motion representation into retargeted motion for the target skeleton through a cross-attention mechanism. The temporal encoder output $\mathbf{H} \in \mathbb{R}^{T \times B \times 24 \times D}$ serves as keys and values for cross-attention, providing the motion context from the source skeleton.

**Decoder input construction.** The decoder input is initialized with random uniform values $\mathbf{Y}_{init} \in \mathbb{R}^{T \times B \times N_t \times D}$, where $N_t$ is the number of joints in the target skeleton. This shape matches the desired output dimensions, allowing the decoder to directly predict joint positions for the target skeleton.

To provide the decoder with essential information about the target skeleton structure and pose, we also incorporate three types of embeddings: (1) Positional embeddings: target-specific positional encodings $\mathbf{E}_{pos}^{(t)} \in \mathbb{R}^{N_t \times D}$ distinguish joint positions within the target skeleton hierarchy. (2) Target joint name embeddings: joint names from the target skeleton are encoded using the same T5 text encoder, producing embeddings $\mathbf{E}_{name}^{(t)} \in \mathbb{R}^{N_t \times D}$ that capture semantic correspondences with source joints. (3) T-pose embeddings: the target skeleton's T-pose configuration $\mathbf{E}_{tpose} \in \mathbb{R}^{N_t \times D}$ provides structural constraints and default joint relationships specific to the target anatomy. Then the enhanced decoder input is formulated as $\mathbf{Y}_{input} = \mathbf{Y}_{init} + \mathbf{E}_{pos}^{(t)} + \mathbf{E}_{name}^{(t)} + \mathbf{E}_{tpose}$.

**Decoder architecture.** The decoder employs a standard transformer decoder architecture with both self-attention and cross-attention mechanisms. The cross-attention mechanism uses the encoder output $\mathbf{H}$ as both keys and values, enabling the decoder to attend to relevant motion patterns from the source skeleton while generating motion for the target skeleton. Then the final decoder output $\mathbf{Y}_{final} \in \mathbb{R}^{T \times B \times N_t \times D}$ represents the predicted retargeted motion, where each position corresponds to a joint trajectory in the target skeleton that preserves the semantic motion content from the source.

The structure of the motion decoder is shown on the right side of Figure. 1.

## 4.4 OVERALL ARCHITECTURE

Figure 2 illustrates the complete pipeline of our cross-skeleton motion retargeting framework when training and testing separately.

For training, given a source motion sequence $\mathcal{M}_A$ performed by skeleton $\mathcal{S}_A$, and a target skeleton $\mathcal{S}_B$, the encoder $E$ takes the source motion and skeleton as input and produces a semantic motion representation: $\mathbf{H_A} = E(\mathcal{M}_A, \mathcal{S}_A)$, where $\mathbf{H_A} \in \mathbb{R}^{T \times B \times 24 \times D}$ captures the essential motion characteristics through our spatial-temporal attention mechanism.

Then the decoder $D$ utilizes both the encoded motion representation $\mathbf{H_A}$ and the target skeleton structure $\mathcal{S}_B$ to generate the retargeted motion $\mathcal{M}_B = D(\mathbf{H_A}, \mathcal{S}_B)$, where $\mathcal{M}_B$ represents the motion sequence adapted for skeleton $\mathcal{S}_B$. Then the predicted motion $\mathcal{M}_B$ and the target skeleton $\mathcal{S}_B$ is sent into the same encoder $E$ to get the semantic motion representation $\mathbf{H_B} = E(\mathcal{M}_B, \mathcal{S}_B)$, where $\mathbf{H_B} \in \mathbb{R}^{T \times B \times 24 \times D}$ captures the essential motion characteristics for motion $\mathcal{M}_B$ and skeleton $\mathcal{S}_B$. Besides that, $\mathbf{H_A}$ and $\mathcal{S}_A$ are also sent into the same decoder $D$ to get predicted motion $\mathcal{M}_{A'}$.

**Motion Reconstruction Training.** To enable the encoder to effectively capture spatial-temporal information from motion sequences conditioned on specific skeleton structures, and to ensure the decoder can accurately decode motion based on attention mechanisms and skeleton constraints, we employ motion reconstruction training. This self-supervised approach trains the model to reconstruct the original motion from its encoded representation.

Specifically, for a given motion-skeleton pair $(\mathcal{M}_A, \mathcal{S}_A)$, we first encode the motion to obtain the semantic representation $\mathbf{H_A}$, then use the same decoder to reconstruct the original motion $\mathcal{M}_{A'} = D(\mathbf{H_A}, \mathcal{S}_A)$.

The reconstruction loss is formulated as a mean squared error between the original motion and the reconstructed motion $\mathcal{L}_{rec} = \text{MSE}(\mathcal{M}_A, \mathcal{M}_{A'}) = \frac{1}{N} \sum_{i=1}^{N} ||\mathcal{M}_A^{(i)} - \mathcal{M}_{A'}^{(i)}||_2^2$, where $N$ represents the total number of joint-frame pairs in the motion sequence. This reconstruction objective ensures that the encoder-decoder pair learns to preserve essential motion information while being conditioned on the specific skeletal structure, providing a strong foundation for cross-skeleton motion transfer.

**Cycle consistency training** While motion reconstruction training ensures the encoder-decoder pair can preserve motion information for a given skeleton, it does not guarantee that the learned representations are skeleton-agnostic. To address this limitation, we introduce cycle consistency training that enforces the encoder to learn motion features that are independent of specific skeleton structures.

The key insight behind cycle consistency training is that if the encoder truly captures skeleton-invariant motion semantics, then the attention representations $\mathbf{H_A}$ and $\mathbf{H_B}$ derived from the same underlying motion should be similar, regardless of the different skeletons on which they are conditioned. By enforcing this consistency, we encourage the encoder to focus on the intrinsic motion characteristics rather than skeleton-specific details.

Specifically, given the attention representations $\mathbf{H_A} = E(\mathcal{M}_A, \mathcal{S}_A)$ from the original motion-skeleton pair and $\mathbf{H_B} = E(\mathcal{M}_B, \mathcal{S}_B)$ from the retargeted motion-skeleton pair, we formulate the cycle consistency loss as $\mathcal{L}_{cyc} = \text{MSE}(\mathbf{H_A}, \mathbf{H_B}) = \frac{1}{T \times B \times 24 \times D} \sum_{t,b,j,d} ||\mathbf{H_A}^{(t,b,j,d)} - \mathbf{H_B}^{(t,b,j,d)}||_2^2$, where the summation is over all temporal frames $t$, batch samples $b$, semantic joints $j$, and feature dimensions $d$.

For testing, our framework performs cross-skeleton motion retargeting in a straightforward forward pass, as illustrated in Figure 2(b). Given a source motion sequence $\mathcal{M}_A$ and target skeleton $\mathcal{S}_B$, the trained encoder $E$ processes the source motion and its corresponding skeleton to extract the skeleton-agnostic motion representation: $\mathbf{H_A} = E(\mathcal{M}_A, \mathcal{S}_A)$. Then, the trained decoder $D$ takes this motion representation along with the target skeleton structure to generate the retargeted motion $\mathcal{M}_B = D(\mathbf{H_A}, \mathcal{S}_B)$.

## 4.5 TRAINING OBJECTIVE.

Our complete training objective combines the reconstruction loss with additional regularization terms to ensure motion quality:

$$\mathcal{L}_{total} = \mathcal{L}_{rec} + \lambda_{cyc}\mathcal{L}_{cyc} + \lambda_{root}\mathcal{L}_{root} + \lambda_{FK}\mathcal{L}_{FK}, \qquad (4)$$

where $\mathcal{L}_{rec}$ and $\mathcal{L}_{cyc}$ are the reconstruction and cycle consistency losses described above, and the additional terms provide crucial constraints for realistic motion generation.

**Root stability loss:** The root stability loss $\mathcal{L}_{root}$ ensures that the global positioning and orientation of the character remain consistent during reconstruction. This loss is computed as the mean squared error between the root position and rotation of the original motion $\mathcal{M}_A$ and the reconstructed motion $\mathcal{M}_{A'}$: $\mathcal{L}_{root} = \text{MSE}(\mathbf{p}_{root}^A, \mathbf{p}_{root}^{A'}) + \text{MSE}(\mathbf{r}_{root}^A, \mathbf{r}_{root}^{A'})$, where $\mathbf{p}_{root}$ and $\mathbf{r}_{root}$ represent the root position and rotation respectively. This constraint is particularly important for maintaining character stability and preventing unrealistic drifting or spinning behaviors.

**Forward kinematic loss:** The forward kinematic loss $\mathcal{L}_{FK}$ enforces anatomical consistency by ensuring that the reconstructed motion respects the skeletal constraints and joint limitations. This loss computes the difference between the forward kinematic solutions of the original and reconstructed motions: $\mathcal{L}_{FK} = \text{MSE}(\text{FK}(\mathcal{M}_A), \text{FK}(\mathcal{M}_{A'}))$, where $\text{FK}(\cdot)$ represents the forward kinematic function that converts joint angles to 3D joint positions.

The hyperparameters $\lambda_{cyc}$, $\lambda_{root}$, and $\lambda_{FK}$ control the relative importance of each loss component. We set $\lambda_{cyc} = 20$, $\lambda_{root} = 7$ and $\lambda_{FK} = 100$ in our experiment.

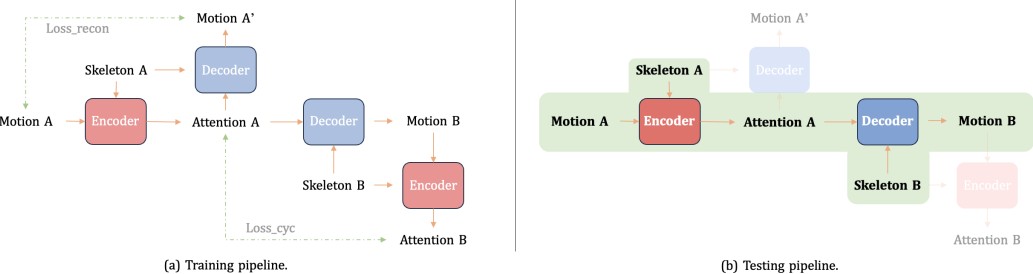

(a) Training pipeline.  (b) Testing pipeline.

Figure 2: The pipeline of our training and testing process.

## 5 EXPERIMENTS

**Implementation details.** Our method is implemented in PyTorch and trained on NVIDIA A100 GPUs. We use the AdamW optimizer with a starting learning rate of $1e-4$, momentum parameters $\beta_1 = 0.9$ and $\beta_2 = 0.99$, and weight decay of $0.999$. The training is conducted with a batch size of 16 and a motion window length of 64 frames.

**Datasets.** We evaluate our method on the Mixamo dataset Adobe (2025). We collect 12 characters (Big Vegas, Warrok W Kurniawan, Michelle, Amy, Castle Guard 01, Doozy, Mousey, Mutant, Prisoner B Styperek, Remy, Goblin, Skeletonzombie T Avelange) for training and 7 characters (Big Vegas, Warrok W Kurniawan, Michelle, AJ, Kaya, Paladin J Nordstrom, Peasant Man) for testing, which in total results in 1756 motion sequences at 30 frames per second. Following previous methods Martinelli et al. (2024); Lee et al. (2023); Zhang et al. (2023); Aberman et al. (2020), we eliminate finger joints from all skeletons to focus on body motion retargeting. However, unlike approaches that select joints by specific joint names, we remove joints by filtering out those containing finger-related identifiers (e.g., "finger", "thumb", "index") and other joints related to human clothes or hair. This deletion-based approach results in skeletons with varying numbers of remaining joints, which better reflects the diversity of real-world skeletal structures and poses additional challenges for cross-skeleton motion transfer. Besides that, we also follow SAN Aberman et al. (2020) to split joints to get more varied skeleton structures with different joint numbers. To comprehensively evaluate our method's generalization capability, we construct four evaluation splits based on the visibility

of characters and motions during training: We have unseen character (uc), unseen motion (um), seen character (sc), and seen motion (sm), resulting in four evaluation scenarios: sc+sm, sc+um, uc+sm, uc+um.

**Baselines.** We compare our method with two recent state-of-the-art approaches for motion retargeting: R$^2$ET Zhang et al. (2023) and PAN Hu et al. (2024). R$^2$ET is a shape-aware method that employs separate networks for skeletal structure and character geometry. Following our problem setting, we train and evaluate only the skeleton network component. Since R$^2$ET is limited to intra-structural motion retargeting, we compare against it only on the same-skeleton scenarios. PAN supports both intra-structural and cross-structural retargeting, enabling comprehensive comparison across all test conditions. For fair evaluation, we train both baselines on our training dataset and evaluate them on our test set using identical experimental protocols. The training dataset contains no paired motion sequences across different characters. We evaluate the retargeted motions by comparing the global joint positions of the generated sequences against the ground truth motions for the target skeleton. The evaluation metric is Mean Squared Error (MSE).

### 5.1 QUANTITATIVE RESULTS

We compare our method with R$^2$ET and PAN in the four evaluation scenarios defined above. The quantitative results are shown in Table 1. Our method demonstrates superior performance across all test conditions, achieving the lowest MSE in both intra-structural and cross-structural motion retargeting tasks. Notably, our approach shows particularly strong improvements in the challenging cross-structural scenarios (us+sm and us+um), where the skeletal differences are most pronounced. The consistent performance gains across different motion types (same vs. unlike motions) indicate that our group-based body part processing and spatio-temporal attention mechanisms effectively capture motion semantics while adapting to diverse skeletal topologies.

| | Intra-Structural | | | | | Cross-Structural | | | | |
|---|---|---|---|---|---|---|---|---|---|---|
| | sc+sm | sc+um | uc+sm | uc+um | mean | sc+sm | sc+um | uc+sm | uc+um | mean |
| R$^2$ET | 0.07388 | 0.15561 | 0.24846 | 0.36392 | 0.21047 | - | - | - | - | - |
| PAN | 0.00629 | 0.00852 | 0.01030 | 0.01079 | 0.00898 | 0.01294 | 0.01412 | 0.01378 | 0.01660 | 0.01436 |
| Ours | **0.00262** | **0.00495** | **0.00297** | **0.00189** | **0.00310** | **0.00282** | **0.00586** | **0.00390** | **0.00226** | **0.00371** |

Table 1: Comparison with state-of-the-art methods on intra-structure and cross-structure motion retargeting. sc: seen character, uc: unseen character, sm: seen motion, um: unseen motion. The evaluation metric is MSE, so a smaller value means a better performance.

### 5.2 QUALITATIVE RESULTS

Figure 3 presents qualitative comparisons of motion retargeting results across different methods. Our approach produces more natural and semantically consistent motions compared to the baselines. The difference in joint count results from our joint deletion policy, in contrast to their joint selection approach compared to their choice policy.

### 5.3 ABLATION STUDY

To validate the effectiveness of each component in our proposed method and understand their individual contributions to motion retargeting performance, we conduct comprehensive ablation studies on key design choices. We systematically analyze the following components: (1) **Group joint policy**: We examine whether allowing shared joints between different body parts improves the grouping strategy compared to strictly disjoint body part assignments. (2) **Joint masking**: We evaluate the impact of randomly masking joints to zero before attention pooling, which is designed to enhance robustness to varying skeletal structures. (3) **Motion representation**: We compare using the full motion representation, including both joint positions and rotations, versus using only rotational information. (4) **Forward kinematics loss**: We analyze whether incorporating the FK loss term improves the physical plausibility and accuracy of the retargeted motions. We also introduce two experiment settings. One is that the training motion set for each character is distinct. The other is that every two characters have at most 10 paired motion sequences. Results shown in Table. 2 indicates that: (1) Allowing shared joints between body parts significantly improves performance

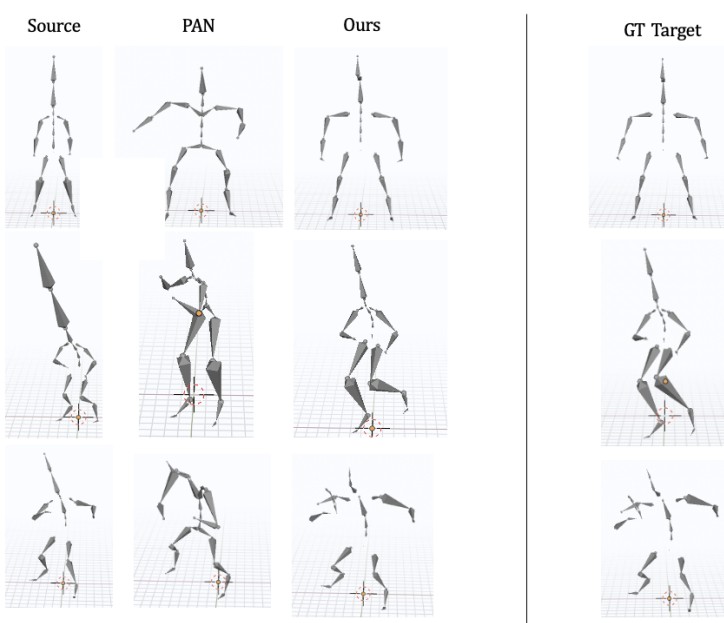

Figure 3: Qualitative results of our method and the two baselines.

across all metrics. Shared joints provide vital connections between different body parts. Without shared connections, body parts can move independently, leading to unnatural motion artifacts and reduced overall quality. (2) The random joint masking strategy demonstrates clear benefits, improving robustness metrics and overall retargeting quality. This validates our hypothesis that masking forces the attention mechanism to learn more generalizable representations that can handle varying skeletal topologies effectively. (3) Including joint positions in the motion representation reduces performance compared to using rotations alone. This occurs because positional information contains skeleton-specific characteristics that make it difficult for the attention pooling network to decompose skeleton structure from pure motion dynamics, hindering the model's ability to generalize across different character structures.

| Distinct | Intra-Structural | Cross-Structural |
|---|---|---|
| w/o share | 0.00428 | 0.00512 |
| w/o mask | 0.00403 | 0.00487 |
| w/ pos | 0.00321 | 0.00394 |
| w/o FK | 0.00346 | 0.00420 |
| Our's Full | 0.00310 | 0.00371 |

| Mix | Intra-Structural | Cross-Structural |
|---|---|---|
| w/o share | 0.00502 | 0.00589 |
| w/o mask | 0.00431 | 0.00501 |
| w/ pos | 0.00344 | 0.00399 |
| w/o FK | 0.00369 | 0.00425 |
| Our's Full | 0.00317 | 0.00422 |

(a) Ablation study for distinct training data.  (b) Ablation study for mixed training data.

Table 2: Ablation study results showing the impact of different components on both intra-structure and cross-structure motion retargeting performance.The performance differences between distinct and mixed training data demonstrate that unpaired data provides the model greater freedom to learn features independent of skeletal structure.

# 6 CONCLUSION

In this paper, we presented a novel skeleton-aware motion retargeting method that effectively transfers motions across characters with different skeletal structures. Our approach leverages spatial attention pooling with joint grouping to extract skeleton-invariant motion representations while preserving essential movement dynamics. We hope this work contributes to advancing cross-character motion transfer techniques and may be beneficial for applications in animation, gaming, and virtual reality where diverse character motions are required.

## A APPENDIX: LLM USAGE

Large Language Models were used to assist with manuscript writing and polishing, specifically for language refinement, grammar checking, and improving text clarity and flow. The LLM was not involved in research ideation, methodology, experimental design, or data analysis—all scientific concepts and contributions were developed by the authors. The authors take full responsibility for all manuscript content and ensure compliance with ethical guidelines.

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
