# OpenReview forum: "SPARK: Spatio-temporal Part-based Attention for Retargeting Cross-skeleton Motion"
_ICLR.cc/2026/Conference — ICLR 2026 Conference Withdrawn Submission_

### Official Review · Reviewer_jbNS · 2025-10-27

**Soundness:** 2
**Presentation:** 1
**Contribution:** 1
**Rating:** 2
**Confidence:** 5

**Summary:**

The paper proposes a learning-based skeletal motion retargeting with per-body-part attention pooling. The six body parts are predefined as torso, left leg, right leg, left arm, right arm, and head. Compared to Hu et al. 2024, these body parts have shared joints on the torso for the holistic coordination of motions. The spatial encoder encodes six body part groups independently with attention pooling to produce 24 tokens (4 tokens per body part). Before the attention pooling, spatial joint information is enriched with the learnable positional encoding, joint name embeddings from the T5 encoder, and T-pose embeddings. The 24 tokens are independently encoded in temporal multi-head attentions to learn motions over time within each body part. The decoder is a standard transformer decoder where the embedding from the encoder is injected into the decoder transformer block as cross-attention. The model is trained with 12 characters from Mixamo with the motion reconstruction loss, cycle consistency loss, root stability loss, and FK loss.

**Strengths:**

The problem of motion retargeting agnostic to skeletal structure makes sense and is well-motivated.

**Weaknesses:**

* Insufficient qualitative evaluation and support
  * No supplementary videos showing the spatio-temporal quality of results
  * Skeletal animations must be checked with skinning, not just with stick figures of skeletons, to make sure there are no undesirable twists
* Questionable core design
  * In theory, a hand-crafted definition of body parts should not be needed if the attention mechanism is working
  * The assumption that motions of one body part should only relate to that body part is debatable. Many motions must respect the relations with other body parts, such as clapping or touching the head. This assumption may have been inspired by previous papers (e.g., Hu et al. 2024), but still good to be critical and rethink what retargeting of animations should really respect.
* Limited dataset
  * A very small subset of Mixamo rigs is insufficient to claim the generalizability to various skeletal structures
  * Mixamo's main body skeletons, excluding fingers and auxiliary joints, are well-standardized
  * The joint names are also relatively consistent in Mixamo. The use of the T5 embedding may be overkill if only Mixamo is used.

**Questions:**

As noted in Weaknesses, please present sufficient visual results for applications in the creative domain. Numerical evaluations should be used to support and explain the visual results, and vice versa. Since the paper presents a spatio-temporal retargeting method, the submission should accompany a video presentation to verify the animation artifacts. Please also present the results with the skinned geometry (which should be easy, as the authors used Mixamo for the data) to convince readers that there are no twisting artifacts. If the ICLR 2026 rebuttal process allows providing additional supplementary videos, please do so.

How diverse is the skeletal structure in the dataset? As far as I know, Mixamo rigs' main body skeletal structure is well-standardized and not a very good dataset to test retargeting scenarios with a different number of joints.

---

### Official Review · Reviewer_WTmv · 2025-10-31

**Soundness:** 3
**Presentation:** 3
**Contribution:** 2
**Rating:** 4
**Confidence:** 3

**Summary:**

In this paper, the authors present a method called Spatio-Temporal Part-based Attention for Retargeting Cross-Skeleton Motion (SPARK) to tackle the problem of motion transfer. The proposed approach introduces a part grouping strategy that enables motion retargeting across skeletons with varying joint numbers, combined with a transformer-based architecture for effective spatio-temporal modeling. Experimental results are provided to demonstrate the advantages of the proposed SPARK method.

**Strengths:**

+ The authors commit to releasing their code to support reproducibility.
+ The motion transfer task is important and has the potential to benefit a wide range of downstream applications.

**Weaknesses:**

+ Missing video comparisons: This submission does not include any video comparisons to demonstrate motion consistency, which is a common evaluation protocol in this field. I strongly encourage the authors to include additional video comparison results against modern baselines to support a more comprehensive and convincing conclusion.
+ Limited experimental validation: Although I am not an expert in motion transfer, the experimental section appears rather limited given the authors’ claim of providing “comprehensive experiments.” Only one figure (Figure 3) is presented for visual evaluation, which is insufficient to assess the method’s performance.
+ Missing related work: Several relevant works are missing from the discussion, which would help contextualize the contribution:
1) Style-ERD: Responsive and Coherent Online Motion Style Transfer, CVPR 2022
2) MCM-LDM: Arbitrary Motion Style Transfer with Multi-condition Motion Latent Diffusion Model, CVPR 2024
3) MoST: Motion Style Transformer between Diverse Action Contents, CVPR 2024

It will be better if the authors add more modern methods as baselines.
+ Lack of efficiency analysis: The authors claim that SPARK significantly reduces training time and computational cost. However, no quantitative evidence is provided to support this claim. I recommend adding a detailed comparison of model efficiency and GPU consumption, similar to Table 2 in Style-ERD.
+ Insufficient evaluation metrics: The paper employs too few quantitative metrics. It would be beneficial for the authors to follow the evaluation settings of MCM-LDM and include metrics such as FMD, CRA, SRA, TSI, and FSF. Extending the ablation studies with these additional metrics would also strengthen the experimental validation.
+ Limited technical novelty: The technical contribution appears to be incremental. The idea of integrating part-based information into a motion transfer framework has already been explored in MoST. Given the limited evaluation and lack of strong comparative evidence, it is difficult to assess the technical effectiveness and originality of the proposed method.

**Questions:**

+ In Figure 3, the leg joints appear disconnected from the torso joints in both the source and ground-truth targets, whereas the PAN results show properly connected skeletons. Could the authors clarify the reason for this discrepancy and provide a more detailed explanation?
+ Could the authors explicitly describe how the qualitative results in Figure 3 should be evaluated? A clearer description of the evaluation criteria would help readers better understand the visual comparisons.
+ It is unclear how the proposed spatio-temporal cross-attention mechanism contributes to improved efficiency. Could the authors elaborate on this aspect and explain the underlying rationale or supporting evidence?

---

### Official Review · Reviewer_bKQa · 2025-10-31

**Soundness:** 3
**Presentation:** 2
**Contribution:** 3
**Rating:** 2
**Confidence:** 5

**Summary:**

A part-based neural network with cross-attention is proposed for motion retargeting between homeomorphic skeletons. The network first group the joint of human body into 6 parts. For skeleton with different number of joints, the joints in each part are first mapped as a fixed number of latent tokens by using cross attention during encoding, by which the motion of skeletons with different joint number are aligned in the latent space. During decoding, the latent joint tokens are quried by tokens, the number of which is the number of the joint of the target skeleton, for motion retargeting. This manuscript uses two types of cross-attention in encoder and decoder to achieve retargeting between skeletons with different number of joint. The proposed method is evaluated on the benchmark dataset Mixamo to show its effectiveness.

**Strengths:**

1. The proposed method achieves good performance compared with the state-of-the-arts;

2. The idea of using cross-attention for motion mapping between skeletons with different joint number is novel and seems reasonable;

**Weaknesses:**

1. The writing of the manuscript is poor.
     * The motivation of using attention pooling for motion mapping between skeletons with different joint number is not clear. What is the difference between SAN, PAN, R2ET,the proposed method and other methods that is designed for the handing of the homeomorphic skeletons retargeting issue? Why the proposed one is better than the other similar methods? I think the main advance is not only the computional efficiency;
     * The definition of some symbols are not clarified. For example in line 190, what is the meaninig of X and tpos?

2. The experiment is not complete. More experiment results are needed to support the claims in the manuscript. For example, the authors claim that the proposed method significantly reduce computational overhead. However there is no experimental result to support it.

3. The number of latent token is fixed as 4 in the manuscript. This design may not effective when the difference of the number of joint between source and target character is large. Meanwhile the subtle motion of some joints may be ignore when the tokens are compressed and represented by only 4 tokens. There should be parameter sensitivity analysis to seed its effectivess.

4. The comparison is not complete. There are many existing methods designed for motion retargeting among homeomorphic skeletons. SAN is one of them. The comparison with the existing method should include SAN.

**Questions:**

1.	What is the influence of the number of latent token mentioned in line 194? Why using 4? There should be parameter sensitivity analysis to support it.

2.	In line 264, is the Y_init learnable? Or is it just randomly initialized in each inference?

3.	How is the closeness between H_A and H_B, especially under different settings of the number of latent token?

**Details Of Ethics Concerns:**

No concern.

---

### Official Review · Reviewer_6fWk · 2025-11-01

**Soundness:** 2
**Presentation:** 1
**Contribution:** 1
**Rating:** 0
**Confidence:** 5

**Summary:**

This paper presents a transformer-based framework for motion retargeting across characters with different skeletal structures. The key challenge addressed is preserving motion semantics while adapting to diverse skeleton topologies.

The proposed method, SPARK, introduces a spatio-temporal, part-based attention mechanism that efficiently models motion across heterogeneous skeletons. Its main components are: Semantic Body Part Grouping, Attention Pooling Encoder, and Cross-Attention Decoder.

Evaluation is performed on Mixamo dataset with 12 training and 7 testing characters. SPARK outperforms R2ET and PAN on MSE.

**Strengths:**

Despite its limitations, the paper demonstrates several strengths across these dimensions:

- Clarity: The paper is well-written and clearly organized, with a logical structure that guides the reader through motivation, methodology, experiments, and conclusions.
- Quality: The authors provide a complete framework that integrates both spatial and temporal attention mechanisms within a part-based motion retargeting system. The method appears to be implemented carefully, and include various loss terms (reconstruction, cycle consistency, root stability, and forward kinematics).

**Weaknesses:**

This paper suffers from several significant weaknesses that limit its originality, experimental rigor, and overall contribution to the field.

## Lack of Novelty and Conceptual Overlap with Prior Work

The proposed SPARK framework is highly similar to existing Skeleton-Aware Networks (SAN, Aberman et al., 2020). The core idea of grouping joints into body parts and performing localized feature aggregation has already been extensively explored in SAN and its follow-ups (e.g., SAME, PAN). The main difference here is that SPARK replaces CNN-based aggregation with attention pooling and a transformer backbone, which is a relatively straightforward architectural substitution rather than a fundamentally new idea. The claimed “spatio-temporal part-based attention” largely mirrors SAN’s hierarchical message passing design, and therefore the conceptual advancement appears incremental and limited in originality.

## Insufficient Experimental Evaluation

The experimental section is notably weak and does not provide convincing empirical evidence to support the method’s advantages.

The baselines are limited to two older methods (R2ET 2023 and PAN 2024), both of which predate more recent motion retargeting or transformer-based approaches such as MeshRet (Ye et al., 2024) and AnyTop (Gat et al., 2025). This narrow comparison prevents a fair assessment of SPARK’s relative performance.

The evaluation metrics are overly simplistic, relying solely on Mean Squared Error (MSE). No perceptual or physically motivated metrics (e.g., motion smoothness, joint angle continuity, or contact accuracy) are considered.

The paper lacks any human subjective evaluation, which is essential in motion quality assessment, since minor geometric errors can lead to perceptually large differences in animation realism.

There is no qualitative or quantitative analysis of how well the retargeted motion drives mesh-based character animation. Showing the retargeted motion on a skeletal stick figure is insufficient to judge plausibility or visual appeal.

## Poor Presentation of Results

The visual results are extremely limited — only one qualitative figure is shown, without animation frames or multi-character comparisons. No supplementary materials (e.g., videos) are provided to illustrate motion quality, which is a major omission for a motion retargeting paper. Without visual evidence, it is difficult to verify the paper’s claims of semantic consistency and motion realism.

**Questions:**

- The proposed semantic body-part grouping and attention pooling mechanisms appear conceptually very close to those in Skeleton-Aware Networks (SAN, Aberman et al., 2020) and Pose-Aware Attention Networks (PAN, Hu et al., 2024). Could the authors clarify the precise conceptual and methodological differences between SPARK and these works? Are there any theoretical or empirical insights suggesting that the part-based attention formulation offers better inductive bias or generalization than SAN/PAN?
- The paper compares only with R2ET (2023) and PAN (2024), which are relatively old. Why were more recent transformer- or diffusion-based motion retargeting methods such as MoMa (Martinelli et al., 2024) or AnyTop (Gat et al., 2025) excluded from comparison?
- The evaluation currently relies solely on MSE, which may not correlate well with perceptual motion quality. Could the authors consider incorporating additional objective metrics, such as smoothness, joint angle deviation, contact stability, or even motion Fréchet Distance (MFD)?
- The paper only includes a single qualitative image without supplementary videos. Since motion realism and temporal coherence are central to this topic, can the authors provide supplementary video comparisons of the retargeted animations?
- Have the authors tested the model on full mesh-driven animations to verify whether the retargeted skeleton motions remain visually plausible when applied to 3D character meshes?

---

### Note · Authors · 2025-11-12

I have read and agree with the venue's withdrawal policy on behalf of myself and my co-authors.